



# Using cellular automata to simulate wildfire propagation and to assist in fire prevention and fighting

Joana G. Freire[1] and Carlos C. DaCamara[1]

[1]Instituto Dom Luiz (IDL), Faculdade de Ciências, Universidade de Lisboa, 1749-016 Lisboa, Portugal

**Correspondence:** Joana Freire (jcfreire@fc.ul.pt)

**Abstract.** Cellular Automata have been successfully applied to simulate the propagation of wildfires with the aim of assisting fire managers in defining fire suppression tactics and in designing fire risk management policies. We present a Cellular Automata designed to simulate a severe wildfire episode that took place in Algarve (southern Portugal) in July 2012. During the episode almost 25 thousand hectares burned and there was an explosive stage between 25 and 33h after the onset. Results obtained show that the explosive stage is adequately modeled when introducing a non-local propagation rule where fire is allowed to spread to the nearest and next nearest cells depending on wind speed. When the rule is introduced deviations in modeled time of burning from estimated time based on hotspots detected from satellite have a root mean square difference of 8.7 hours for a simulation period of 46h (less than 20%). The simulated pattern of probabilities of burning as estimated from an ensemble of 100 simulations show a marked decrease out of the limits of the observed scar, indicating that the model represents an added value for fire fighting in what respects to the choice of locations to allocate resources for fire combat.

## 1 Introduction

Wildfires in the Mediterranean region have severe damaging effects that are mainly caused by large fire events (Amraoui et al., 2013, 2015). Restricting to Portugal, wildfires have burned over 1.4 million hectares in the last decade (Sá and Pereira, 2011), and the recent tragic events caused by the megafires of June and October 2017 have left a deep mark at the political, social, economic and environmental levels. Given the increasing trend in both extent and severity of wildfires (Pereira et al., 2005; Trigo et al., 2005; Pereira et al., 2013; DaCamara et al., 2014; Panisset et al., 2017), the availability of modeling tools of fire episodes is of crucial importance. Two main types of models are generally available, the so-called deterministic and stochastic models. Deterministic models attempt a physics-based description of fires, fuel and atmosphere as multiphase continua prescribing mass, momentum and energy conservation, which typically leads to systems of coupled partial differential equations to be solved numerically on a grid. Simplified descriptions, such as FARSITE (Finney, 2004) neglect the interaction with the atmosphere and propagation of the fire front is made using wave techniques. Cellular Automata (CA) are one of the most important stochastic models (Sullivan, 2009); space is discretized into cells, and physical quantities take on a finite set of values at each cell. Cells evolve in discrete time according to a set of transition rules, and the states of the neighboring cells.

CA models for wildfire simulation fill a gap between deterministic and empirical models (Rothermel, 1972, 1983), as they prescribe local, microscopic interactions typically defined on a square (Clarke et al., 1994) or hexagonal (Trunfio, 2004)





grid. The complex macroscopic fire spread dynamics is simulated as a stochastic process, where the propagation of the fire front to neighboring cells is modelled via a probabilistic approach. CA models directly incorporate spatial heterogeneity in topography, fuel characteristics and meteorological conditions, and they can easily accommodate any empirical or theoretical fire propagation mechanism, even complex ones (Collin et al., 2011). CA models can also be coupled with existing forest fire models to ensure better time accuracy of forest fire spread (Rui et al., 2018). More elaborated CA models that overcome typical constraints imposed by the lattice (Trunfio et al., 2011; Ghisu et al., 2015) perform comparably to deterministic models such as FARSITE, however at a higher computational cost.

In the present work, we set up a simple and fast CA model designed to simulate wildfires in Portugal. As benchmark, we have chosen the CA model developed by Alexandridis et al. (2008, 2011) that presents the advantage of having been successfully applied to other Mediterranean ecosystems, namely to the propagation of historical fires in Greece to simulate fire suppression tactics and to design and implement fire risk management policies. This model further offers the possibility to run a very high number of simulations in a short amount of time, and is easily modified by implementing additional variables and different rules for the evolution of the fire spread.

We then present and discuss the application of the CA model to the "Tavira wildfire" episode in which approximately $24,800$ ha were burned in Algarve, a province located at the southern coast of Portugal. The event took place in summer 2012, between July 18 and 21, and fire spread in the municipalities of Tavira and São Brás de Alportel. The Tavira wildfire was one of the largest fires in recent years (excluding the megaevents of the last fire season of 2017), and most of the variables (e.g. total burned area, time to extinction) are well documented and available from official authorities. This fire event was also studied by Pinto et al. (2016), providing a suitable setup for testing the CA model. In addition, comparing the simulation results to this baseline scenario allowed us to identify and formulate the most promising model modifications and refinements to be incorporated in the simulation algorithm.

This paper is organized as follows. A description of the fire event to be modeled and of all data required for simulation and validation of results is provided in Section 2. The rationale behind the setting up of the Cellular Automata is described in Section 3. Results are presented and discussed in Section 4, paying special attention to the modeled temporal and spatial deviations from results derived from location and time of detection of hotspots as identified from remote sensing. Conclusions are drawn in Section 5.

## 2 Data description and processing

### 2.1 The fire event of Tavira

As mentioned in the introduction, we apply a CA model to a large and well documented wildfire that occurred in July 2012 in the Tavira and São Brás de Alportel municipalities, located in Algarve, Portugal (Figure 1). The fire was first reported on July 18 (at about 13h UTC) and was considered as contained on July 21 (at about 17h UTC). The fire burned approximately $24,800$ ha, mainly shrublands which made up about $64\%$ of the affected area, and spread in heterogeneous, predominantly





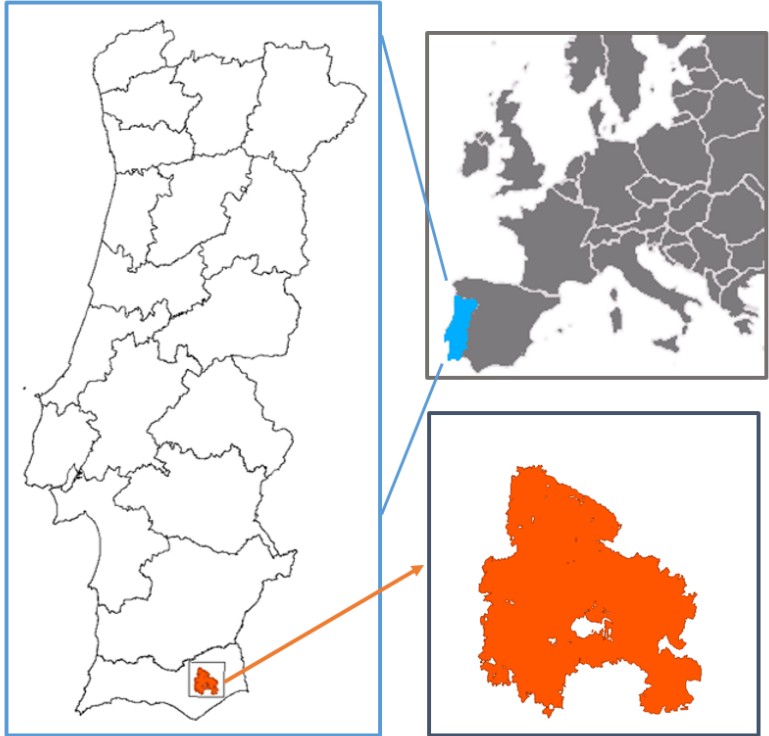

**Figure 1.** Left panel: map of Portugal with the location of the Tavira wildfire, where orange represents the burned scar and the black box indicates the study area used in the simulations. Right panels: schematic representation of Europe with Portugal highlighted in blue (top panel) and zoom of the study area (bottom panel).

steep terrain. It was the largest wildfire in Portugal in 2012, contributing to more than $22\%$ of the total amount of $110,232$ ha of burned area (ICNF, 2012) in that year.

The terrain is prevailingly steep, with slopes of $20\%$ located in the higher altitude region in the northern part of the Tavira municipality with hilltops reaching up to $541$ meters. The altitude and slope decrease towards the southeast area of Tavira and southwest area of the São Brás de Alportel municipalities, having slopes between $0$ and $20\%$ and lower altitudes reaching sea level at several locations (Viegas et al., 2012).

Since 2012 was a year of extreme drought, the meteorological background conditions were very prone to the occurrence of large fire events (Trigo et al., 2013). The region of Tavira is characterized by Mediterranean climate, the maximum monthly temperature in August ranging from $25^oC$ to more than $30^oC$, with maximum absolute temperatures around $39^oC$, and mean relative humidity below $65\%$ (ANPC, 2012). In 2012, the precipitation in Tavira was $45\%$ below the normal record and the study area had a soil water content value below $10\%$ at the time of the fire (Viegas et al., 2012). The wildfire propensity was further aggravated by above average precipitation in 2010 and 2011, that favored vegetation growth and fuel build up.



Consequently, fire danger as measured by the Canadian Fire Weather Index (FWI) System (Wagner, 1974, 1987; Pinto et al., 2018) was rated Extreme with FWI reaching 56.7 during the fire period (Viegas et al., 2012).

The fire propagated in two distinct phases. In the first stage, from 13:00 UTC on July 18 to 17:00 UTC on July 19, the fire burned about 5,000 ha, representing one fifth of the total burned area. In this phase, the fire advanced through rugged

terrain, and wind direction was highly variable, causing frequent shifts in the direction of maximum spread, which was mainly towards south/southeast until it reached the Leiteijo stream, where it gained speed under the influence of topography. Then, in a transition stage, around 16:30 UTC on July 19, the fire started spreading through steep slopes along the Odeleite stream. Spotting occurred up to hundreds of meters due to low fuel moisture, and multiple spot fires were recorded. Fire suppression was difficult due to the steepness of terrain and frequent wind direction changes, and operations were focused on life and

property salvation.

In the second stage from 17:00 UTC to 24:00 UTC on July 19 the fire turned into a major conflagration, greatly increasing its propagation speed and burning about 20,000 ha in 7 hours.

When the fire reached the Odeleite stream it became orographically channeled, as an increase in wind speed led to fast and intense fire growth towards south, where copious amounts of fuel loading were present. The fire split into two advanced

sections heading west and east to the São Brás de Alportel and the Tavira municipalities, with a 10 km wide fire front. In addition, spotting now created new fires up to two kilometers ahead of the fire front. All these factors allowed rapid propagation of the fire front while turning suppression extremely difficult.

## 2.2   Input data

A study area of 30 km x 30 km was defined centered on the burned area (Figure 1) and fine-scaled raster data from various

sources were collected and pre-processed in a common format suitable as input for the wildfire simulations. Data include the ignition points, the start and end times of the fire event, the fire perimeters, the burned areas, the surface wind speed and direction, the topography, and information about the landcover (vegetation type, vegetation density, areas burnt in previous wildfires, waterlines and roads).

Patch-slope information was derived from elevation data as obtained from the Digital Elevation Model provided by the

Shuttle Radar Topography Mission (Farr et al., 2007).

Hourly wind data were obtained from a regional weather simulation performed with the Weather Research and Forecast model (WRF), version 3.1.1 (Skamarock et al., 2008). The quality of the simulation was previously assessed for wind (Cardoso et al., 2012; Soares et al., 2014). WindNinja (version 2.1.3) (Forthofer, 2007) was then used to spatially model the hourly wind input data taking into account the interaction with topography. The temporal behavior of the wind field was then validated

against the information contained in the report by the Portuguese Authority for Nature and Forest (ICNF) (ICNF, 2012).

Fuel type and density were derived by combining information from CORINE Land Cover raster maps at 100 m resolution (CLC2006), the National Forest Inventories produced by ICNF, and the MODIS-based annual Maximum Green Vegetation Fraction (Broxton et al., 2014). Vegetation types were aggregated into four main categories: areas without vegetation, cultivated areas, shrubs and forests (Figure 2, top left panel). The density of vegetation was also stratified into four categories: areas

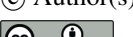


**Table 1.** Assigned values to probability factors of vegetation type ($p_{veg}$) and density ($p_{dens}$).

| categories | $p_{veg}$ | | categories | $p_{dens}$ |
|---|---|---|---|---|
| no vegetation | -1 | | no vegetation | -1 |
| cultivated | -0.4 | | sparse | -0.3 |
| forests | 0.4 | | normal | 0 |
| shrub | 0.4 | | dense | 0.3 |

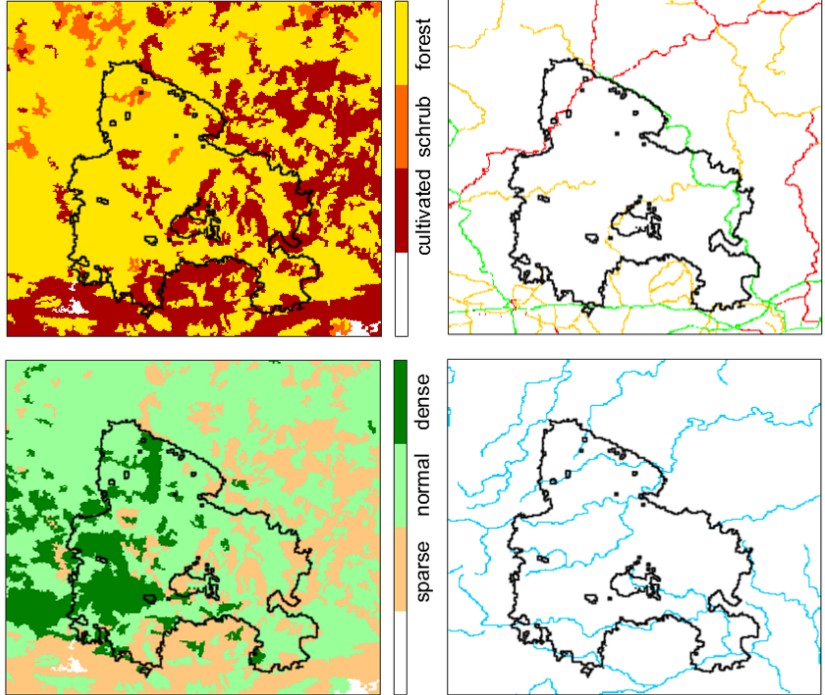

**Figure 2.** Left column: vegetation type (top) and density classes (bottom) inside the study area as indicated by the discrete colorbars. White corresponds to areas without vegetation. Right column: the roads (top) and waterlines (bottom) identified inside the simulation area. Primary, secondary and tertiary roads are represented, respectively, in red, orange and green. Waterlines are all colored in blue.

without vegetation, areas with sparse, normal and dense vegetation (Figure 2, bottom left panel). As described in section 3.1, for the different categories of vegetation type and density, values of the associated probability factors, respectively $p_{veg}$ and $p_{den}$, were empirically assigned or taken from literature (Alexandridis et al., 2008). Assigned values are listed in Table 1. Roads and waterlines inside the simulation area (Figure 2, top right panels) were also included in the model by assigning low

5    values to both probability factors $p_{veg}$ and $p_{dens}$, with $p_{veg} = p_{dens}$. Primary, secondary and tertiary roads were assigned the values $-0.8, -0.7$ and $-0.4$, respectively, whereas the value of $-0.4$ was assigned to the waterlines.



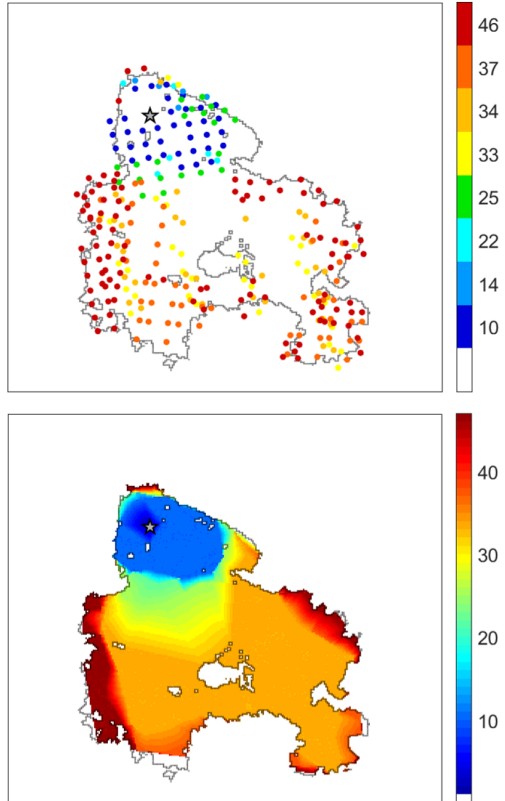

**Figure 3.** Centroids of the active fires detected by MODIS (top panel) and derived times of burning for the cells inside the burned scar (bottom panel). Colors of the centroids and of the cells represent the elapsed time (in hours) since the beginning of the fire event as indicated in the colorbars. The star represents the fire ignition point reported and the black line the perimeter of the burned area.

Active fire data as identified from satellite were used for the quality assessment of the CA model simulations by evaluating temporal and spatial discrepancies between active fire observations and simulated fire growth. For this purpose, we used the MODIS (MODerate Resolution Imaging Spectroradiometer) active fire product that provides hotspots detected at 1km × 1km pixel resolution, at the time of the satellite overpass. The MODIS sensor on the Terra and Aqua satellites supply daytime and nighttime observations at four nominal acquisition times, thus providing information about the geographical location, date, and time of the detected active fires (Giglio et al., 2003).

For each satellite overpass totally or partially covering the total burned area by the Tavira fire, we used the centroids of the active fire footprints (Figure 3, top panel) to define a polygon confined to the burn scar. Times of burning of cells inside the fire scar were then estimated by bilinearly interpolating between the outer limits of the defined polygons (Figure 3, bottom panel).





## 3 Methods

### 3.1 Baseline model

Simulations by the reference CA model developed by Alexandridis et al. (2008) make use of a square grid with propagation to the 8 nearest and next-nearest neighbors (Figure 4). Each cell (or site) is characterized by 4 possible discrete states, corresponding to burning, with fuel not-yet burned, fuel-free, and completely burned cells. The model has 4 possible rules of

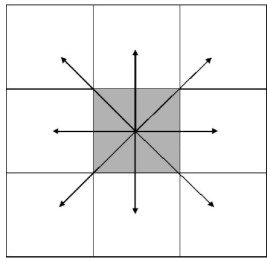

**Figure 4.** The 8 possible fire spread directions on the square grid.

evolution which take into account fuel properties, wind conditions, and topography. The rules are applied at each time step and are described as follows: Rule 1: a cell that cannot be burned stays the same; Rule 2: a cell that is burning down at present time will be completely burned in the next time step; Rule 3: a burned cell cannot burn again; Rule 4: if a cell is burning down at the present time and there are next-nearest neighbor cells containing vegetation fuel, then the fire can propagate to its neighbors

10 with a probability $p_{burn}$ which is a function of the variables that affect fire spread.

In its basic formulation, probability $p_{burn}$ of a given cell depends: 1) on a constant reference probability that a cell in the neighborhood of a burning cell (containing a given type of vegetation and density) starts burning at the next time step under no wind and flat terrain, $p_0$; 2) on the vegetation type, $p_{veg}$, and vegetation density, $p_{den}$; 3) on topography, $p_s$; and 4) on wind fields, $p_w$, as follows:

$$p_{burn} = p_0(1 + p_{veg})(1 + p_{den})p_w p_s. \tag{1}$$

As described in section 2.2, in order to account for the effect of vegetation, both type and density were stratified into discrete classes, and for each class a constant probability factor was assigned as specified in Table 1.

The effect of the wind is modeled as:

$$p_w = \exp[V(c_1 + c_2(\cos(\theta) - 1))], \tag{2}$$

20 where $c_1$ and $c_2$ are adjustable coefficients, $V$ is the wind speed, and $\theta$ is the angle between the wind direction and the fire propagation direction. As expected, $p_w$ increases when wind and fire directions are aligned.

The probability factor that models the effect of the terrain elevation is given by:

$$p_s = \exp(a_s \theta_s), \tag{3}$$





where $\theta_s$ is the slope angle of the terrain and $a_s$ is a coefficient that can be adjusted from experimental data. Slope angle $\theta_s$ was derived from elevation data, $E$, according to

$$\theta_s = atan[(E_1 - E_2)/D], \tag{4}$$

where $D$ is equal to the size $L$ of the square cell when the two neighboring cells are adjacent or to $\sqrt{2}L$ when the two cells are
diagonal. As expected this topography effect is higher when the fire spreads uphill.

## 3.2 Modified model

In order to better mirror the role played by the wind in fire propagation, a modification was introduced in the model by means of a simple new non-local propagation rule. In contrast with the baseline rule $N_1$ that at each time step fire can only spread to its nearest and next-nearest neighbors (Figure 5), according to the new rule $N_2$, depending on given wind velocity thresholds,
the neighborhood affected by the fire propagation is increased in the wind direction, i.e., the number of ignitable cells (or sites) increases (from one up to 10 sites) in the direction of the wind as the wind speed increases.

The model with the new propagation rule $N_2$ will be hereafter referred to as the modified model.

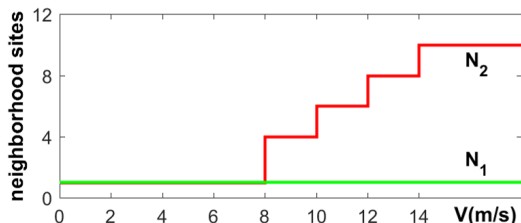

**Figure 5.** The baseline fire spread rule ($N_1$) and the new wind propagation rule ($N_2$).

## 4 Results

### 4.1 Simulations

The landscape was discretized into square cells with size of 100 m and the model free parameters were set according to Alexandridis et al. (2008), i.e. with $p_0 = 0.58$, $a_s = 0.078$, $c_1 = 0.045$ and $c_2 = 0.131$. The time step of the model was set by performing 100 simulations of the propagation of fire inside the observed burned area under no-wind and flat-terrain conditions. The time step was then estimated by dividing the observed time from the starting ignition up to fire containment (46h) by the mean number of time steps required to burn the entire area. The obtained time step was about 20 minutes.
The fire event was then modeled using a probabilistic approach based on ensembles of 100 model simulations. The probability that a given cell burns is accordingly estimated by the fraction of runs where that cell was modeled as a burned one. It is worth emphasizing that, in all runs, burning was confined to the observed burned area by means of an appropriate setting of




the model parameters along the boundary of the final observed scar. However, this setting along the scar boundary is not an artificial device since it reflects the known a posteriori fact that the shape of the scar resulted from effective fire combat in these locations.

Two different ensembles of 100 simulations were generated, one with the baseline fire spread model and the other with the
modified model. Results obtained at four selected stages of the fire are displayed in Figure 6. When using the baseline rule (Figure 6, left panels), and excepting for the slot at 25h (after ignition) where there is a fair agreement between the simulated burned area and the front lines of the fire as indicated by the hotspots identified by satellite, the simulated burning is well behind the fire front, an indication that the modeled propagation of the fire is too slow. A strong contrast is observed when using the modified model (Figure 6, right panels). In this case, the modeled burned areas spread much closer to the fire front
as defined by the hotspots. The exception is the slot at 25h (after ignition), where the modeled propagation of fire is faster than the one suggested by the location of the hotspots. On the other hand, it is worth emphasizing that the explosive behavior of fire between slots at 25h and at 33h is very well simulated when using the new wind propagation rule.

## 4.2   Quality assessment

Burned area in each one of the two ensembles was identified by assuming that a given pixel is a burned one when the modeled
probability that it burned is larger than $0.2$. Each pixel identified as burned was assigned the respective time step as an indicator of the modeled time of burning (Figure 7, left panels). Time deviations were then computed (Figure 7, right panels) by subtracting the times of burning as derived from the hotspots identified by MODIS (Figure 3, bottom panel). Results obtained are consistent with those of the previous section. When using the baseline wind rule (Figure 7, upper panels), the model shows a progressive delay in the propagation of fire, the isochrones of fire propagation attaining values larger that 46h well before the
fire front reaches the southern boundary of the scar. This delay reflects in the positive values of the deviations of modeled time of burning from the one derived from satellite observation and it is worth noting that the delay takes place during the explosive stage of the fire between 25 and 33h (Figure 3, lower panel). When using the modified model (Figure 7, lower panels) the explosive stage of the fire is much better modeled albeit a too fast propagation of the fire front during the first stage. This behavior is reflected in the deviations that present negative values during the first 12h and much lower positive values than the
baseline model between 25 and 33h, an indication that the modified model tends to be closer to the observations than the baseline model. The overall behavior of both models is well summarized by the respective values of bias and of root mean square differences: the $4.4$h ($-6.4$h) of the baseline model (modified model) is consistent with the too slow (too fast) propagation of the modeled fire fronts whereas the root mean square difference of $10.9$h ($8.7$h) points to a behavior closer to observations of the modified model.
The improved behavior of the modified model when compared with the baseline model is also revealed when analyzing the fraction of burned pixels of the scar in successive periods of 6h (Figure 8). The most conspicuous feature is the burning of more than $50\%$ of the total amount of burned cells between 30 and 36h. This explosive stage of the fire is completely missed by the baseline model, whereas the burned area by the modified model reaches $35\%$. When the fraction of burned area estimated from remote-sensed hotspots is small (between 0 and 6h, 12 and 18h, 18 and 24h) both models tend to overestimate that fraction,





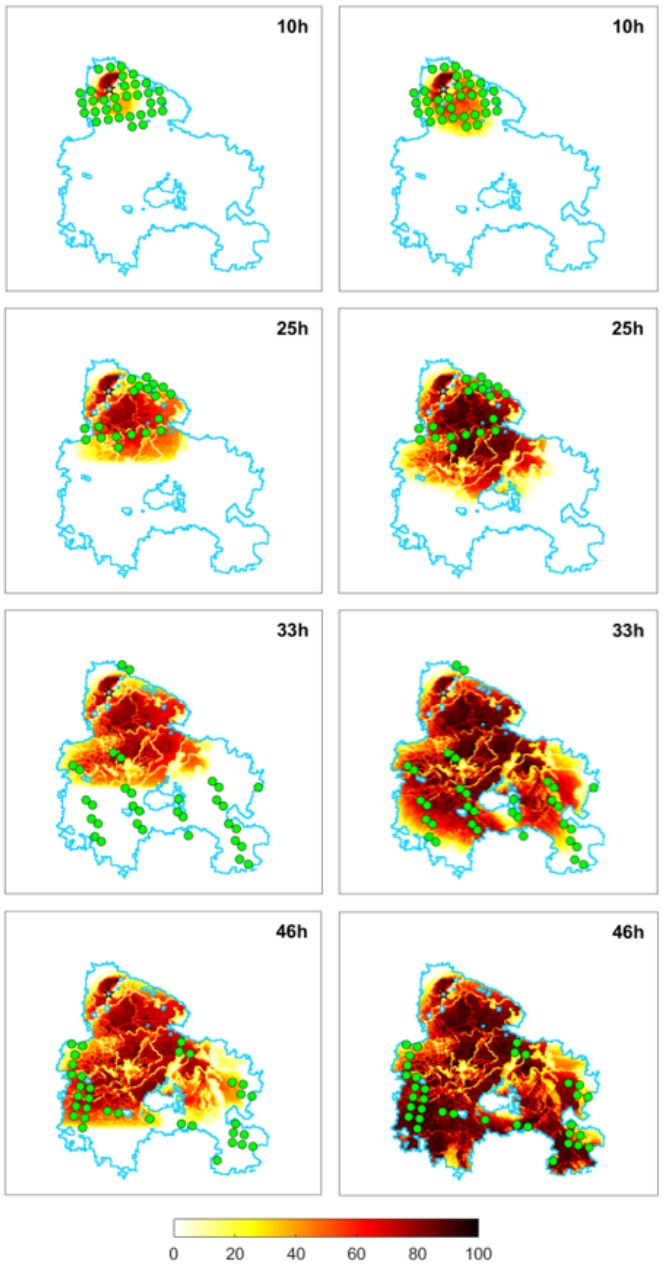

**Figure 6.** Probabilities of burning (%) for the baseline model (left) and for the modified model (right). Colors represent the percentage of burned cells as indicated by the color bar in the bottom of the figure and white represents unburned cells. The star locates the fire ignition point, the blue line the perimeter of the burned area, and the green circles represent active fires as detected by MODIS. Both simulations were restricted to the burned area.



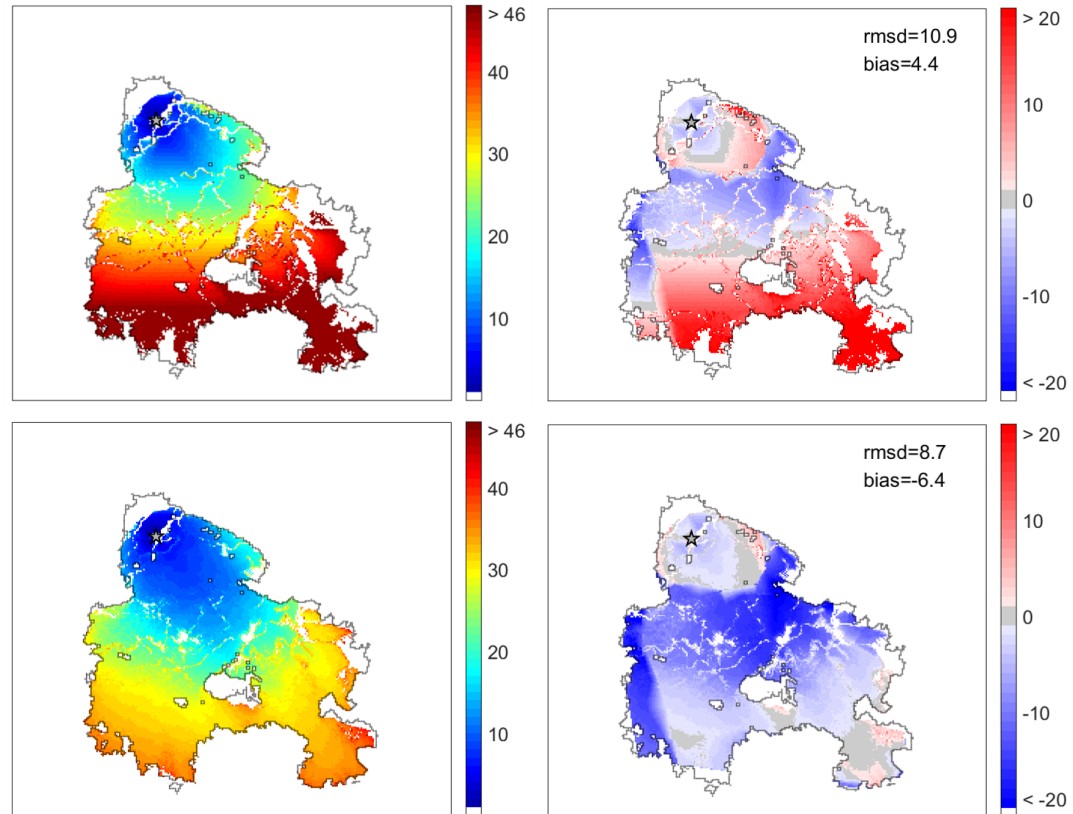

**Figure 7.** Left panels: fire propagation using a threshold of 0.2 for probability of burning for a set of 100 random simulations of the baseline model confined to the burned area (top panel) and of the modified model (bottom panel). Colors represent the elapsed time in hours after the fire ignites. Right panels: time deviations from the left panels relative to the active fires detected by MODIS. Red (blue) shading corresponds to a progressive delay (advance) in fire propagation observed in the CA model, and light gray to an agreement between the CA model and the MODIS active fires. The star represents the fire ignition point, the black line the perimeter of the burned area and white represents unburned cells.

especially the modified model. Between 6 and 12h, the burned fraction simulated by the modified model is close to the burned area estimated from hotspots, whereas the baseline model underestimates that fraction. An opposite behavior occurs in the last interval, between 36 and 42h, where the fraction simulated by the baseline is close to the fraction estimated from hotspots and the modified model underestimates that fraction.

## 4.3   Unconstrained runs

A third ensemble of 100 model simulations by the modified model was performed, this time with no other constraints besides the lattice domain, i.e., in which all fire propagation simulations stop by themselves. The obtained pattern of burning probabilities (Figure 9, left panel) shows a marked decrease outside the limits of the burned scar and this may be put into evidence by



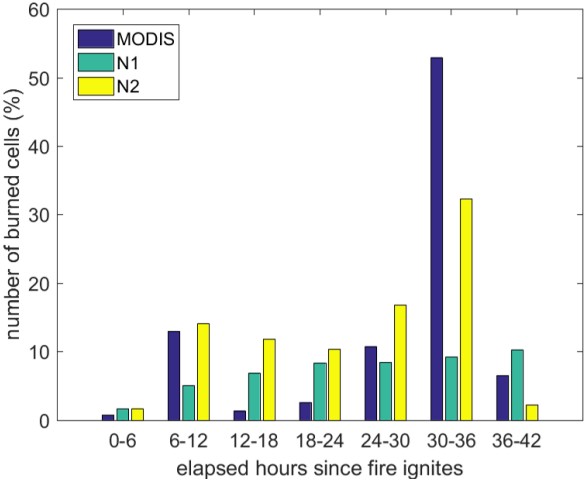

**Figure 8.** Percentage of the total number of burned cells as derived from active fires (MODIS), the baseline model (N1) and the modified model (N2). Each triplet of columns corresponds to the burned cells identified, respectively, in the intervals [0,6[, [6,12[, [12,18[, [18,24[, [24,30[, [30,36[ and [36,42[ hours.

restricting the burned area to cells with a burning probability larger than 80% (Figure 9, right panel). Since fire containment was mainly due to actions by firemen along the perimeter, results indicate that unconstrained simulations represent a very useful tool to assist decision makers during a fire event, by providing indications about locations of low burning probability to be selected as appropriate to allocate resources for fire combat.

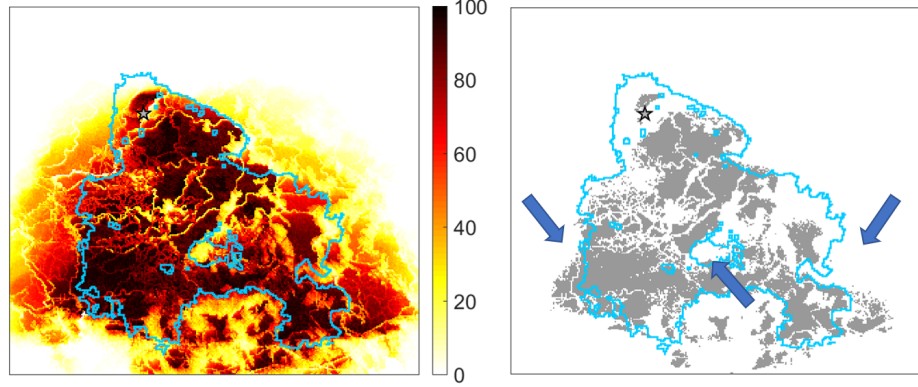

**Figure 9.** Left panel: probabilities of burning (%) as derived from an ensemble of 100 unconstrained simulations of the modified model. The colorbar indicates the probability of burning. The star represents the fire ignition point, the black line the perimeter of the burned area and white represents unburned cells. Right panel: burned sites identified above a threshold probability of 80%. The blue arrows indicate locations where fire combat occurred, namely along the lateral sides of the burned scar and populated areas .





## 5  Conclusions

Cellular Automata (CA) are useful tools in simulating fire dynamics and they are unique in their flexibility to the introduction of changes in properties of individual cells as well as of transition rules. This characteristic together with the low computational cost they require, makes CAs especially useful for fire management, either when planning controlled fires or when making

decisions about combat in an operational scenario. In this paper we set up a CA model designed to respond to wind-driven wildfires. The model is applied to a large wildfire event that took place in southern Portugal in July 2012. Besides its relevance in terms of burned area that reached almost $25,000$ hectares, the event turned into a major conflagration between 25 and 33h after the onset, because of orographic channeling accompanied by an increase in wind speed. This explosive stage represented an ideal scenario to test a CA model designed for wind-driven wildfires. The simulation of the wildfire propagation was made

using a probabilistic approach based on ensembles of 100 simulations that allow estimating the probability of burning of a given cell by the fraction of runs where that cell was modeled as a burned one. Two different ensembles were generated, based on two models that were analogous except in the wind propagation rule. In the baseline model, the wind rule only applied to the nearest and next-nearest neighbors, whereas in the modified model the neighborhood affected increased with wind speed, reaching up to 10 cells. Results obtained pointed to a progressive delay in the propagation of fire simulated by the baseline

model that contrasted with a moderate advance obtained with the simulations by the modified model. The contrast in overall behavior of the two ensembles reflects in the obtained values of bias and root mean square deviations between simulated times of burning of each cell and respective estimated times from hotspots as identified by remote sensing. The value of $4.4h$ ($-6.4h$) for bias in the case of the baseline (modified) model indicates the overall delay (advance) of the simulation and the improved performance of the modified model is suggested by the value of $8.7h$ of the root mean square difference that is more than two

hours lower than the value of $10.9h$ obtained with the baseline model. Differences between the two ensembles are conspicuous during the explosive stage of the wildfire when more than $50\%$ of the area burned between $30-36h$ after the fire onset. The baseline model simulated less than $10\%$ of the area whereas the modified model reached $35\%$. The usefulness of the modified model as a tool to assist fire managers in locating resources for firefighting during a fire event was tested by performing a third ensemble of 100 simulations in which the fire propagation is unconstrained, and the simulations stop by themselves. Results

obtained show a marked decrease of probability of burning outside the observed fire scar, in good agreement with the effective placement of the allocated combat forces during the real event. The proposed CA model with a non-local wind rule presented a very good temporal and spatial performance and revealed potential to be an added value in fire management. Currently the model is being tested in different scenarios, namely with the very large fire events that took place in Portugal in June and October 2017. The transition rules that were used in the CA model do not consider neither the state of stress of vegetation

nor the meteorological conditions. In line with Alexandridis et al. (2011), incorporation of these two aspects is currently being considered by associating probability factors to the Drought Code (DC) and to the Fine Fuel Moisture Code (FFMC), two indices of the Canadian Fire Weather Index System that respectively provide a numerical rating of seasonal drought effects and of the ease of ignition and the flammability of fine fuel at the daily level.



*Author contributions.* JGF performed the study. JGF and CDC wrote, discussed the results and reviewed the manuscript.

*Competing interests.* The authors declare that they have no competing interests

*Acknowledgements.* JGF was supported by the Post-Doctoral grant SFRH/BPD/101760/2014 from FCT, Portugal. Part of this work was also supported by the project BrFLAS - Brazilian Fire-Land-Atmosphere System (FAPESP/1389/2014) funded by national funds through the
5  Portuguese Foundation for Science and Technology (FCT). Part of the data was provided by the FIRE-MODSAT project, financed by FCT, Portugal (Contract EXPL/AGR-FOR/0488/2013).





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
