# Peer review of "Using cellular automata to simulate wildfire propagation and to assist in fire management"

_Natural Hazards and Earth System Sciences, 2018_

## Referee Comment (RC1) · Anonymous Referee #1 · 3 Dec 2018

This paper presents an application of a cellular automata model to simulate wildfire propagation. The CA modeling is presented as a useful tool to support decision-makers during fire events, providing information about the location of future fire spread and allowing them to design a proper solution to reduce the propagation of fires.

The paper is well structured and well written. However, I would suggest the authors do an effort to slightly reduce the length of section 2.1 in order to further improve the readability of the manuscript.

From the methodological point of view, the paper proposes an advance of the model proposed by Alexandridis in 2008. In the modified model, a stronger relevance is given

to the role of wind speed in fire propagation. The proposed model seems to give better results, at least for the case study presented in the paper.

My only concerns are related to the model uncertainty. All the results are presented in terms of probability. The latter results from the ensemble of 100 models run. It would be of extreme interest to map model uncertainty; without any information about it, it would be very difficult to use the proposed model as a decision-making support tool. About the relevance of uncertainty there is a number of relevant papers in literature; as a first reading, I suggest Fischhoff and Davis 2014. Communicating scientific uncertainty. Proceedings of the National Academy of Sciences 111: 13664-13671. Moreover, it would be interesting to have a sensitivity analysis concerning the variation of certain a priori fixed parameters, as the $c_1$, $c_2$ and as a coefficient of the model (which are now settled based on the values proposed by Alexandridis). Similarly, it could be interesting to further explore the sensitivity of the result to the choice of the 0.2 probability threshold applied in section 4.2.

---

## Referee Comment (RC2) · Anonymous Referee #2 · 4 Dec 2018

General comments

The manuscript is an interesting exercise whereby the authors try to replicate the fire growth patterns of a large wildfire in Portugal. I find some problems regarding the structure of the work:

- Section 2 is essentially methods, so I suggest moving it to the methods section.

- Sections 4.2 and 4.3 are a mix of results and discussion. Reorganizing it and renaming it "Discussion" would be an improvement and it could be the place to discuss the shortcomings indicated below.

[Figure]

- The Conclusion is too long. I suggest to rename it "Summary and Conclusion".

The ms. suffers from lack of explanation/rational for a number of decisions made regarding modeling, which seem too arbitrary. There is some confusion regarding the concept of probability, which in this ms. is (at least in part) akin to fire spread rate. Also, better discussion is needed of the strengths and weaknesses of the approach, especially given that I was not impressed with its performance and the tuning procedures seem to apply only to this fire in particular.

Title: Replace "fire prevention and fighting" by "fire management, more encompassing and elegant.

Specific comments

P2, L17-20. This sentence is far from accurate. Current fire spread models and tools used operationally are deterministic but are empirical in nature, e.g. FARSITE. Thus the usage of deterministic models that "attempt a physics-based description of fires, fuel and atmosphere as multiphase continua prescribing mass, momentum and energy conservation, which typically leads to systems of coupled partial differential equations" is restricted to the realm of research, as they are impossible to use in real time and their outputs are erratic. In fact, the physical mechanisms of fire spread remain largely unknown.

P2, L24. Again, empirical models are deterministic. "fill a gap" suggests they are somewhat half-way between empirical and physics-based models, which is not true. They are simply of a different kind.

P3, L3. I would better describe terrain as "undulated" rather than "steep".

P3, L11. Better use "fire proneness" instead of "wildfire propensity".

P3, L12. This statement is too strong. Antecedent years rainfall influences subsequent fire activity in fuel-limited systems, which is not really the case of Portugal.

P4, L1. "Consequently" indicates that the previous information leads to this conclusion. However, fire danger as per the FWI is determined by a combination of atmospheric influences and fuel dryness, being independent of fuel accumulation. The previous sentence refers drought and the typical weather conditions in the area, but not the prevailing weather during or before the fire, namely the dominant effect of wind speed. Revise (also, "consequently" should not initiate a paragraph).

P4, L10. "protection", not "salvation".

P4, L14. Replace "copious . . . loading" by "heavy fuel loads".

P4, L34. "shrubland", not "shrubs". Also, better use farmland or agriculture than culti-vated, as the latter is ambiguous (it can denote planted forests).

P5, Table 1 and subsequent: you need to state what "probability" refers to. Fire spread?

P7, L10. From what follows, probabilities p express relative rate of spread rather than the probability of fire spread, e.g. according to a rule commonly used, rate of spread of the headfire perimeter is one order of magnitude faster than the backfire rate of spread, but their likelihood of spread is the same. Please make this more explicit.

P7, L22. This in fact is the effect of slope, not the effect of elevation.

P8, L8. You need to explain the rational for this rule and where does it come from. Physically it does not make sense, unless the increase in flame size due to wind would extend to new cells, which is impossible to happen.

P8, L15. The first two paragraphs in Results, and part of the 3rd, belong in Methods.

P8, L16-19. If I understood correctly, this parameterization and the obtained time step depends on this wildfire rate of spread, right? Consequently, it cannot be applied to fires with different rate of spread.

P9, L2. Also, it depends of the changes in fuel types, or is this already accounted for in more effective fire fighting? (use fighting or suppression, not combat - replace here

and elsewhere in the ms.).

P9, L15. Why 0.2? A threshold of p=0.5 is usually assumed for go/no-go events. Again, this initial paragraph belongs to Methods.

P11, L6. Methods.

P12. "Since fire containment was mainly due to actions by firemen along the perimeter". You don't know if this was the case (also in Fig. 9 legend) ... in may instances it should be due to changes in fuel type, topographic effects, or the presence of linear interruptions such as roads.

P13, L22. Still, 35% implies a high degree of underestimation in fire growth.

P13, L25. I don't see how this connection can be made, because probability of burning in this model is independent of fire suppression operations, and I don't think you know how fire suppression operations were carried out and where the resources were placed.

P13, L26. Why is this rule non-local? It suggests it is somewhat universal but no rational or basis was given for the rule.

P13, L27. I don't think there's evidence for this "very good" performance. Also, the added value for fire management is not proven. Why is this model preferable to fire growth simulators that are becoming more and more used operationally? Indicate advantages and disadvantages (preferably not here but previously in the Discussion).

---

## Referee Comment (RC3) · Anonymous Referee #3 · 10 Dec 2018

The manuscript represents an interesting contribution on the field of stochastic cellular automata models for wildfire propagation and it's well structured and written. The improvement over the reference model can be of great interest for operational use. Following are some comments regarding the manuscript.

- In the introduction, the authors wrote that deterministic models attempt to a physic-based description of process. As a matter of fact, several deterministic empirical models exist and are widely used operationally (e.g. FARSITE). Please change the formulation of this section.

- In section 3.2 the description of the modified neighborhood rule is given. This should

be the most detailed part of the method section, since it represents the major innovation over the baseline model. However, I've found the description to be lacking important details. The authors wrote that the fire propagation neighborhood is extended in the wind direction, but it's unclear how they consider wind directions that are not aligned with the possible propagation directions on the 2d lattice of the grid (e.g. directions that are not multiple of 45°). Are you considering all the cells in the N2 neighborhood or only the boundaries of the region? How is the neighborhood shaped? Does the shape depend on wind speed? Please extend the description in order to include more details.

- In section 4.1, the methodology for assessing the time step used by the model is explained. In my opinion, it's important to point out that the method used to estimate the time step cannot be used during operational activities, and this represents a major throwback in the actual applicability of the model in real field usage. Model-wise, it's also important to note the limit of using a fixed time step for the propagation of fire on a fixed lattice, hence implying a fixed rate of spread. Following these considerations, the analysis of the performances of the model regarding the propagation time assessment are not very relevant. Please justify your modeling choices or include some considerations on this issue in the discussion.

---

## Author Comment (AC1) · 29 Dec 2018

The authors thank Referee #1 for his very constructive comments.

1. The paper is well structured and well written. However, I would suggest the authors do an effort to slightly reduce the length of section 2.1 in order to further improve the readability of the manuscript.

Answer: As suggested by the referee this Section is now substantially reduced and reads as follows: As mentioned in the introduction, we apply a CA model to a large and well documented wildfire that occurred in July 2012 in the Tavira and São Brás

de Alportel municipalities, located in Algarve, Portugal (Figure 1). The fire was first reported on July 18 (at about 13h UTC) and was considered as contained on July 21 (at about 17h UTC). The fire burned approximately 24,800 ha, mainly shrublands which made up about 64% of the affected area, and spread in heterogeneous, undulated terrain. It was the largest wildfire in Portugal in 2012, contributing to more than 22% of the total amount of 110,232 ha of burned area (ICNF, 2012) in that year. Since 2012 was a year of extreme drought, the meteorological background conditions were very prone to the occurrence of large fire events (Trigo et al., 2013). The fire propagated in two distinct phases. In the first stage, from 13:00 UTC on July 18 to 17:00 UTC on July 19, the fire burned about 5;000 ha, representing one fifth of the total burned area. In this phase, the wind direction was highly variable and the fire advanced through rugged terrain, with frequent shifts in the direction of maximum spread until it reached the Leiteijo stream. In the second stage from 17:00 UTC to 24:00 UTC on July 19 the fire turned into a major conflagration, greatly increasing its propagation speed and burning about 20,000 ha in 7 hours. When the fire reached the Odeleite stream it became orographically channeled, as an increase in wind speed led to fast and intense fire growth towards south, where heavy fuel loads were present. The fire split into two advanced sections heading west and east to the São Brás de Alportel and the Tavira municipalities, with a 10 km wide fire front. In addition, spotting created new fires up to two kilometers ahead of the fire front. All these factors allowed rapid propagation of the fire front while turning suppression extremely difficult.

2. The latter results from the ensemble of 100 models run. It would be of extreme interest to map model uncertainty; without any information about it, it would be very difficult to use the proposed model as a decision-making support tool.

Answer: The reviewer points out a relevant issue. However, we consider that mapping model uncertainty would be beyond the scope of a feasibility study such as the one we are describing. However, the issue of uncertainty is now discussed at the end of the new Section "5. Summary and conclusion", where the following sentences

are added: Finally, it may be noted that results from the CA models are presented in terms of probability of burning as an outcome of ensembles of runs. This raises the issue of providing information of model uncertainty that is especially relevant if the CA model is to be used as a decision-making support tool. As discussed in Fischhoff and Davis (2014), characterizing model uncertainty involves identifying key outcomes, characterizing variability as well as internal and external validity, and finally summarizing uncertainty. Presentation of the impacts on fraction of burned area, bias and root mean square deviations when choosing different thresholds of probability of burning are a first step towards conveying results of uncertainty. Further steps in this direction will have to involve direct contacts with decision-makers when analyzing other large fire events namely the above-mentioned ones that took place in Portugal in June and October 2017.

3. Moreover, it would be interesting to have a sensitivity analysis concerning the variation of certain a priori fixed parameters, as the $c_1$, $c_2$ and as a coefficient of the model (which are now settled based on the values proposed by Alexandridis)

Answer: A sensitivity analysis to parameters $c_1$ and $c_2$ is now included in the manuscript (Figure 1) in the new subsection "2.5 Simulations" (of the new section "2 Data and methods"). The following sentences and figures were added to the manuscript: A sensitivity study was also performed to assess the effects of constants $c_1$ and $c_2$ on the propagation of fire (Equation 2). As shown in Figure 6, simulated values of total burned area and of burned area inside the perimeter of the fire scar increase (decrease) with increasing $c_1$ (increasing $c_2$). Moreover, above (below) a certain threshold of $c_1$ ($c_2$), a progressive departure is observed between the simulated values of total burned area and of burned area inside the perimeter of the fire scar, an indication that the simulated fire is spreading out of the recorded limits. Choice of $c_1$ = 0.045 and $c_2$ = 0.131 (Alexandridis et al., 2008) represents a compromise between burning a large fraction of the area inside the perimeter and spreading a small fraction outside.

Figure 6: Simulated values of the total burned area (red curves) and of the burned area inside the perimeter of the fire scar (blue curves) in units of the total area inside the perimeter as a function of $c_1$ for fixed $c_2 = 0.131$ (left panel) and as a function of $c_2$ for a fixed $c_1 = 0.045$ (right panel).

4. Similarly, it could be interesting to further explore the sensitivity of the result to the choice of the 0.2 probability threshold applied in section 4.2.

Answer: A sensitivity analysis to the choice of probability thresholds is now included in the manuscript (Figure 2) in new subsection "3.1 Constrained runs" (of new section "3 Results"). The following sentences and figures were added to the manuscript: Burned area in each one of the two ensembles was identified by assuming that a given pixel is a burned one when the modeled probability that it burned is larger than a fixed threshold. Each pixel identified as burned was assigned the respective time step as an indicator of the modeled time of burning. Time deviations were then computed by subtracting the times of burning as derived from the hotspots identified by MODIS (Figure 3, bottom panel). Finally, three measures of quality of the simulations were derived for different thresholds of probability, namely the fraction of burned area (relative to the total area inside the perimeter of the fire scar), the bias (simulated time minus time derived from hotspots) and root mean squared differences (between simulated time and time derived from hotspots). Figure 8 presents results obtained when using the model with the baseline wind rule (dashed lines) and the modified model (solid lines). In both cases, and as to be expected, the fraction of burned area decreases with increasing values of the threshold (Figure 8, top panel), the baseline model always presenting, for each threshold, lower values of burned area than the modified model. The baseline (modified) model presents positive (negative) values of bias for each threshold (Figure 8, middle panel) meaning that, on average, the simulations are late (in advance) when compared with times derived from satellite. In both cases, the bias increases with increasing values of threshold, the baseline model becoming more and more biased and the modified model approaching zero bias, although the rate of increase is smaller

than the one of the baseline model. Finally, the root mean square difference (Figure 8, bottom panel) shows an opposite behaviour in the two cases, with values increasing (decreasing) with the threshold in the case of the baseline (modified) model. When considering all together the three measures of quality of the simulations, the modified is better performant than the baseline model and choosing values of threshold between 0.4 and 0.6 represents a good compromise in terms of simulated burned area and simulated time of fire propagation.

Figure 8: Fraction of the burned area inside the perimeter relative to the total area inside the perimeter of the fire scar (top panel), bias (middle panel) and root mean square difference (bottom panel) as a function of the probability threshold for $c_1$ = 0.045 and $c_2$ = 0.131. The dashed lines correspond to the baseline model and the solid lines to the modified model.

[Figure]

**Fig. 1.** Figure 6

**Fig. 2.** Figure 8

---

## Author Comment (AC2) · 29 Dec 2018

The authors thank Reviewer #2 for his positive comments that greatly contributed to improving the manuscript.

General comments

1. Section 2 is essentially methods, so I suggest moving it to the methods section.

Answer: Section 2 is now part of the new "Section 2. Data and methods".

2. Sections 4.2 and 4.3 are a mix of results and discussion. Reorganizing it and renaming it "Discussion" would be an improvement and it could be the place to discuss

the shortcomings indicated below.

Answer: Sections 4.2 and 4.3 were reorganized as suggested and the manuscript now contains a new "Section 3. Results" and a new "Section 4. Discussion".

3. The ms. suffers from lack of explanation/rational for a number of decisions made regarding modeling, which seem too arbitrary. There is some confusion regarding the concept of probability, which in this ms. is (at least in part) akin to fire spread rate. Also, better discussion is needed of the strengths and weaknesses of the approach, especially given that I was not impressed with its performance and the tuning procedures seem to apply only to this fire in particular.

Answer: We agree with the reviewer about the misleading use of the term "probability factor", namely in sections 2.2 and 3.1 of the original manuscript. We used the term "probability factor" to be consistent with Alexandridis et al. (2008) who used the term "probability". Since "probability factors" do not in fact represent probabilities, we have replaced the term "probability factors" by "loadings".

4. The Conclusion is too long. I suggest to rename it "Summary and Conclusion".

Answer: The new Section 5 was renamed as suggested.

5. Title: Replace "fire prevention and fighting" by "fire management, more encompassing and elegant.

Answer: The title was changed as suggested.

Specific comments

6. P2, L17-20. This sentence is far from accurate. Current fire spread models and tools used operationally are deterministic but are empirical in nature, e.g. FARSITE. Thus the usage of deterministic models that "attempt a physics-based description of fires, fuel and atmosphere as multiphase continua prescribing mass, momentum and energy conservation, which typically leads to systems of coupled partial differential equations"

is restricted to the realm of research, as they are impossible to use in real time and their outputs are erratic. In fact, the physical mechanisms of fire spread remain largely unknown.

Answer: We agree with the reviewer and the text now reads as follows: Wildfire propagation is described in a variety of ways, be it the type of modelling (deterministic, stochastic), type of mathematical formulation (continuum, grid-based) or type of propagation (nearest-neighbor, Huygens wavelets), and often the formulation adopted combines different approaches (Sullivan, 2009; Alexandridis et al., 2011). For instance, the classic model of Rothermel (1972, 1983) combines fire spread modeling with empirical observations, and simplified descriptions such as FARSITE (Finney, 2004) neglect the interaction with the atmosphere and the fire front is propagated using wavelet techniques. Cellular Automata (CA) are one of the most important stochastic models (Sullivan, 2009); space is discretized into cells, and physical quantities take on a finite set of values at each cell. Cells evolve in discrete time according to a set of transition rules, and the states of the neighboring cells.

7. P2, L24. Again, empirical models are deterministic. "fill a gap" suggests they are somewhat half-way between empirical and physics-based models, which is not true. They are simply of a different kind.

Answer: We agree with the reviewer and the sentence "fill a gap between deterministic and empirical models" was removed. The entire paragraph now reads as follows: CA models for wildfire simulation prescribe local, microscopic interactions typically defined on a square (Clarke et al., 1994) or hexagonal (Trunfio, 2004) grid. The complex macroscopic fire spread dynamics is simulated as a stochastic process, where the propagation of the fire front to neighboring cells is modeled via a probabilistic approach. CA models directly incorporate spatial heterogeneity in topography, fuel characteristics and meteorological conditions, and they can easily accommodate any empirical or theoretical fire propagation mechanism, even complex ones (Collin et al., 2011). CA models can also be coupled with existing forest fire models to ensure better time accuracy of

forest fire spread (Rui et al., 2018). More elaborated CA models that overcome typical constraints imposed by the lattice (Trunfio et al., 2011; Ghisu et al., 2015) perform comparably to deterministic models such as FARSITE, however at a higher computational cost.

8. P3, L3. I would better describe terrain as "undulated" rather than "steep".

Answer: The text was changed accordingly.

9. P3, L11. Better use "fire proneness" instead of "wildfire propensity".

Answer: As suggested by Referee #1, section 2.1 was substantially reduced (see answer to comment #1 by Referee #1). This sentence was accordingly removed from the manuscript.

10. P3, L12. This statement is too strong. Antecedent years rainfall influences subsequent fire activity in fuel-limited systems, which is not really the case of Portugal.

Answer: We agree with the reviewer. The sentence was removed from the manuscript (see also answer to comment #9).

11. P4, L1. "Consequently" indicates that the previous information leads to this conclusion. However, fire danger as per the FWI is determined by a combination of atmospheric influences and fuel dryness, being independent of fuel accumulation. The previous sentence refers drought and the typical weather conditions in the area, but not the prevailing weather during or before the fire, namely the dominant effect of wind speed. Revise (also, "consequently" should not initiate a paragraph).

Answer: We agree with the reviewer. The sentence was removed from the manuscript (see also answer to comment #9).

12. P4, L10. "protection", not "salvation".

Answer: The sentence was removed from the manuscript (see also answer to comment #9).

13. P4, L14. Replace "copious : : : loading" by "heavy fuel loads".

Answer: The text was changed accordingly.

14. P4, L34. "shrubland", not "shrubs". Also, better use farmland or agriculture than cultivated, as the latter is ambiguous (it can denote planted forests).

Answer: "Shrubs" and "cultivated" were replaced by "shrubland" and "agriculture", respectively.

15. P5, Table 1 and subsequent: you need to state what "probability" refers to. Fire spread?

Answer: The term "probability factors" was replaced by "loadings" (see also answer to comment #3).

16. P7, L10. From what follows, probabilities p express relative rate of spread rather than the probability of fire spread, e.g. according to a rule commonly used, rate of spread of the headfire perimeter is one order of magnitude faster than the backfire rate of spread, but their likelihood of spread is the same. Please make this more explicit.

Answer: In fact, as shown in Eq. 2, this effect is taken into account by the wind loading (pw), that makes the probability of backpropagation one magnitude lower than the frontal propagation. For instance, making c1=0.045, c2=0.131, and V=10m/s, we have loading pw equal to 1.5683 for theta=0° and equal to 0.1142 for theta=180°.

17. P7, L22. This in fact is the effect of slope, not the effect of elevation.

Answer: "Elevation" was replaced by "slope".

18. P8, L8. You need to explain the rational for this rule and where does it come from. Physically it does not make sense, unless the increase in flame size due to wind would extend to new cells, which is impossible to happen.

Answer: This new rule intends to incorporate the effects due to fire spotting that

were reported during the second stage of the Tavira event, when the wind speed was stronger. This is now explicitly mentioned in the manuscript: In order to better mirror the role played by the wind in fire propagation, a modification was introduced in the model by means of a new rule that allows propagation to non-adjacent cells with the aim of incorporating the effects due to fire spotting.

19. P8, L15. The first two paragraphs in Results, and part of the 3rd, belong in Methods.

Answer: The two paragraphs are now part of the new subsection 2.5 Simulations" (in new section 2. Data and methods").

20. P8, L16-19. If I understood correctly, this parameterization and the obtained time step depends on this wildfire rate of spread, right? Consequently, it cannot be applied to fires with different rate of spread.

Answer: Yes, the reviewer is correct, but this paper is a feasibility study. As mentioned in the manuscript, currently, we are applying the same model to other fire events, namely the events in Portugal on 15 October 2017 (where wind played a crucial role). Results obtained so far indicate good performance when these fire events are simulated with the proposed model and time steps.

21. P9, L2. Also, it depends of the changes in fuel types, or is this already accounted for in more effective fire fighting? (use fighting or suppression, not combat - replace here.

Answer: Yes, the reviewer is correct, and the sentence (in new subsection "2.5 Simulations") now reads: It may be noted that this setting along the scar boundary is not an artificial device since it reflects the known a posteriori fact that the shape of the scar resulted from effective fire-fighting in locations where changes in fuel types and the presence of roads make fire propagation harder.

22. P9, L15. Why 0.2? A threshold of p=0.5 is usually assumed for go/no-go events.

Again, this initial paragraph belongs to Methods.

Answer: We agree with the reviewer's point of view and results are now presented for threshold p=0.5.

23. P11, L6. Methods.

Answer: The text was moved to the new subsection 2.5 Simulations" (in new section 2. Data and methods").

24. P12. "Since fire containment was mainly due to actions by firemen along the perimeter". You don't know if this was the case (also in Fig. 9 legend): : : in may instances it should be due to changes in fuel type, topographic effects, or the presence of linear interruptions such as roads.

Answer: The phrasing is in fact misleading. We just wanted to point out that results of the unconstrained simulations indicate that the probability of burning is lower beyond the actual perimeter of the fire scar (as a result of changes in fuel type, topographic effects and the presence of linear interruptions such as roads). Since (successful) actions by firemen took place mostly along the perimeter of the fire scar, unconstrained simulations are a useful tool to assist decision makers during a fire event, by providing indications about locations of low burning probability to allocate resources for firefighting. Incidentally, information about fire combat were obtained from the fire report and from Rui Almeida (personal communication), who works at the National Forest Institute (ICNF) and took part in the fire fighting. The point raised by the reviewer is now explicitly mentioned in the new subsection "3.2 Unconstrained runs": Unconstrained simulations therefore indicate that the probability of burning is lower beyond the actual perimeter of the fire scar as a result of changes in fuel type, topographic effects and the presence of linear interruptions such as roads.

25. P13, L22. Still, 35% implies a high degree of underestimation in fire growth.

Answer: We are referring to 35% out of 55% of the area that burned during the explosive stage. It is true that there is still an underestimation, but this result is to be compared with the value of 10% that is obtained when using the baseline wind rule. The text was slightly changed to better reflect the above-mentioned point.

26. P13, L25. I don't see how this connection can be made, because probability of burning in this model is independent of fire suppression operations, and I don't think you know how fire suppression operations were carried out and where the resources were placed.

Answer: Results obtained show a marked decrease of probability of burning outside the observed fire scar, suggesting that this type of model may help decision-makers about the placement of the allocate fire-fighting forces during a fire event.

27. P13, L26. Why is this rule non-local? It suggests it is somewhat universal but no rational or basis was given for the rule.

Answer: "Non-local" was used in the sense that fire was allowed to propagate to non-adjacent cells. The sentence now reads: The proposed CA model with a wind rule that allows fire propagation to non-adjacent cells represents an improvement to the baseline model and reveals potential to be an added value in fire management.

28. P13, L27. I don't think there's evidence for this "very good" performance. Also, the added value for fire management is not proven. Why is this model preferable to fire growth simulators that are becoming more and more used operationally? Indicate advantages and disadvantages (preferably not here but previously in the Discussion).

Answer: The following sentences were added at the end of the new "Section 4 Discussion": The flexibility to the introduction of changes in properties of individual cells (e.g. when imposing constraints to fire propagation along the perimeter of the fire scar) as well as of transition rules (e.g. the proposed one on the effects of wind), together with the required low computational cost (that allows performing a very large number of runs in a short amount of time) make CA adequate tools to be used, either when planning

controlled fires or when making decisions about fighting in an operational scenario. For instance, we are currently developing a mobile application (app) that allows the user to run the proposed modified model over the study area and modify the properties of the individual cells.

---

## Author Comment (AC3) · 29 Dec 2018

The authors thank Referee #3 for his very constructive comments.

1. In the introduction, the authors wrote that deterministic models attempt to a physic based description of process. As a matter of fact, several deterministic empirical models exist and are widely used operationally (e.g. FARSITE). Please change the formulation of this section.

Answer: We agree with the Reviewer and the text about fire propagation models was substantially revised (see answers to Comments #6 and #7 by Referee #2).

[Figure]

2. In section 3.2 the description of the modified neighborhood rule is given. This should be the most detailed part of the method section, since it represents the major innovation over the baseline model. However, I've found the description to be lacking important details. The authors wrote that the fire propagation neighborhood is extended in the wind direction, but it's unclear how they consider wind directions that are not aligned with the possible propagation directions on the 2d lattice of the grid (e.g. directions that are not multiple of 45). Are you considering all the cells in the N2 neighborhood or only the boundaries of the region? How is the neighborhood shaped? Does the shape depend on wind speed? Please extend the description in order to include more details.

Answer: We agree with the reviewer and the text was expanded as follows: In order to better mirror the role played by the wind in fire propagation, a modification was introduced in the model by means of a new rule that allows propagation to non-adjacent cells with the aim of incorporating the effects due to fire spotting. In contrast with the baseline rule N1 that at each time step fire can only spread to its nearest and next-nearest neighbors, according to the new rule N2, for each burning cell at a given time step, fire propagation is modeled according to the two following steps: apply the baseline wind rule and determine the direction(s) of fire spread (if any) for each cell in the next-nearest neighborhood. If: i) according to the previous step, the fire propagates to a new cell, ii) the wind speed at the considered burning cell is above the threshold of 8 m/s and iii) the angle between the wind direction and the displacement vector (from the considered burning cell to the newly ignited cell) is lower than pi/10, then fire also spreads to a number of other contiguous cells (along the displacement vector), the number of ignited cells depending on the wind speed at the considered burning cell (Figure 5). The model with the new propagation rule N2 will be hereafter referred to as the modified model.

3. In section 4.1, the methodology for assessing the time step used by the model is explained. In my opinion, it's important to point out that the method used to estimate the time step cannot be used during operational activities, and this represents a major
throwback in the actual applicability of the model in real field usage. Model-wise, it's also important to note the limit of using a fixed time step for the propagation of fire on a fixed lattice, hence implying a fixed rate of spread. Following these considerations, the analysis of the performances of the model regarding the propagation time assessment are not very relevant. Please justify your modeling choices or include some considerations on this issue in the discussion.

Answer: The point raised by the reviewer is relevant since fair estimates of the time step are required given that the inputs of wind information require adequate temporal information. The new wind propagation rule that we are proposing is an attempt to circumvent the problem of having a fixed time-step for the propagation of fire on a fixed lattice. This is now mentioned in the Conclusion section.